# Digital Transformation of Land Services in Indonesia: A Readiness Assessment

**Kusmiarto Kusmiarto [1,2,*]**, **Trias Aditya [1]**, **Djurdjani Djurdjani [1]** and **Subaryono Subaryono [1]**

[1] Department of Geodetic Engineering, Faculty of Engineering, Universitas Gadjah Mada, Yogyakarta 55281, Indonesia; triasaditya@ugm.ac.id (T.A.); djurdjani@ugm.ac.id (D.D.); ssubar@ugm.ac.id (S.S.)

[2] Sekolah Tinggi Pertanahan Nasional, Yogyakarta 55293, Indonesia

[*] Correspondence: kusmiarto@stpn.ac.id; Tel.: +62-274-550987

**Abstract:** In 2020, digital transformation was a major theme to commemorate Indonesia's main agrarian law's anniversary. This theme is a reminder of the need to fully implement digital services to improve the quality of land registration products that are cheap, easy to operate, perform quickly, and are trusted by the community. However, no research has comprehensively assessed the readiness of the digital transformation of land services in Indonesia. This paper aims to evaluate the readiness of a land office to achieve digital transformation visions. Here, we apply the Digital Governance Assessment Framework (DGRA), adapted to the land service sector, as the basis for conducting this evaluation. The nine core indicators of the DGRA toolkit are used as a basis for assessment. Desk studies were conducted to identify formal legislation and to find the technical specifications. Direct observations and in-depth interviews were conducted with stakeholders to find user needs and evaluate the implementation of current regulations on the land service business process. Quality assessment was carried out on land registration data at the Land Office of Yogyakarta City as a sample. The quality assessment results indicate a problem with completeness, conformity, consistency, accuracy, duplication, and integrity. In conclusion, the readiness level still needs improvement, especially in the indicator related to Cyber Security, Privacy, and Resilience (1.0). Even though the Leadership and Governance, User-Centered Design, and Public Administration Reforms and Change Management sections shows a reasonably high score (≥2.0), other core sections, namely Technology Infrastructure (1.7), Legislation and Regulation (1.4), Data Infrastructure, Strategies, and Governance (1.8) are mediocre, and therefore they need improvement.

**Keywords:** digital land governance; smart cadastres; digital land services; digital governance readiness assessment; land administration services; Indonesia

## 1. Introduction

Today's digital transformation of land administration is a necessity. Land offices must carry this out to provide efficient land services. Services that are transparent, fast, inexpensive, easy to perform, and produce reliable products to guarantee legal certainty within a modern land administration system based on the cadastral engine, are needed to achieve the sustainable development goals (SDGs) related to land [1,2]. Land services need to be implemented effectively. They must be fit for purpose, appropriate and adequate, interoperable and sustainable, flexible and inclusive, and able to accelerate efforts to document, record, and recognize community relations in all their forms [3]. It will not be easy to provide efficient and effective land services without utilizing information technology. Although modernization in land services is expensive and takes a long time, the decision to start is the right one [4]. Digital transformation with information technology in the land sector can enhance the economic maturity level, helping to realize sustainable development goals. It also affects the economic growth rate [5]; a 10% increase in internet

access correlates with an increase of Gross Domestic Product (GDP) by 1.35% in developing countries and by 1.19% in developed countries [6].

Innovations in the land administration system can increase the economy and ease of doing business in a country [7]. Many countries are carrying out the digital transformation of their land administration services. However, they face challenges in realizing a good land system. According to international indicators, for example, based on the World Bank index [8], one of the biggest challenges includes the availability of a land administration system with high reliability, transparency, coverage, levels of conflict, and accessibility to land rights. If these indicators can be achieved, then the legal certainty of land rights (rights security) can be delivered to landowners, automatically increasing the land market's efficiency [8]. These indicators are based on the Ease of Doing Business (EoDB) assessment, a ranking system established by the World Bank Group [8]. In the EoDB index, a higher rating indicates better business regulation, usually simpler terms (easy, cheap, effective, and efficient), and stronger protection of property rights. Digital transformation in registering property may be essential for a country to achieve faster and more efficient land services.

Currently, the Ministry of Agrarian and Spatial Planning/National Land Agency (ATR/BPN), a government agency that provides land services in Indonesia, is preparing to implement digital transformation in all land service processes. The Ministry of ATR/BPN has launched four types of electronic-based land services and implemented them nationally: Electronic Mortgages, Land Value Zone Information (ZNT), Land Certificate Check, and Land Registration Information Letter (SKPT) [9]. There are at least 72 types of land services in land offices in Indonesia, which are grouped into six service groups: First Time Land Registration Services, Land Registration Data Maintenance Services, Land Registration, and Information Services, Land Measurement Services, Land Regulation, and Arrangement Services, and Complaint Management Services [10]. The challenge is to ensure that all types of land services can be implemented digitally.

A study related to the implementation of fully digital land registration activities has been published. This research is on the quality and usability assessment of information technology in the context of participatory land registration in Indonesia (PaLaR) [11]. PaLaR is a research pilot project to implement fully digital land registration activities, including collecting, processing, presenting, and storing land registration data. Collecting and recording data on a mobile application, both physical and juridical data are carried out by the Community Land Registration Committee/community representatives (CLRC). The study of PaLaR has provided a method of implementing a full digital land registration service based on community participation. However, the PaLaR study has not specifically assessed the government readiness if full digital registration is implemented nationally.

Many frameworks for evaluating land service effectiveness have been published, one of which is the evaluation framework for assessing the urban cadastral system's reliability in Ethiopia [12]. This framework comprehensively proposes the basis for evaluating urban cadastral system policies that refer to and synthesize many other standard frameworks. They are EFQM [13], Cadastre 2014 [14], Land Governance Assessment Framework (LGAF) [15], Evaluation of Land Administration Systems [16], Systematic Evaluation of Land Administration Systems [17], The Cadastral Template 2.0 [18], and The 2030 Agenda for SDGs [19]. The framework provides indicators of an ideal urban cadastral system. These indicators can be used for the systematic evaluation of the cadastral system. Aspects related to legal, institutional, economic, social, political, environmental, technical, and public-private partnerships are combined as a consideration in this framework. Unfortunately, the framework has not specifically discussed assessing digital transformation readiness in the land service sector.

Another study related to the performance evaluation of land services in urban areas (Addis Ababa, Ethiopia) has been conducted [20]. The evaluation is based on the European Foundation for Quality Management (EFQM) model [13]. The variables used as a reference for evaluation are nine criteria in the EFQM model. The evaluation method is carried out through focus group discussions, interviews, and questionnaires. In this regard, the

exploration is new in that there is no investigation led in the study area utilizing the EFQM framework. However, it has not specifically discussed the readiness level of implementing digital cadastral systems in urban areas.

The World Bank has developed a framework for assessing digital governance readiness, namely the Digital Governance Readiness Assessment (DGRA) framework [21]. The DGRA is a comprehensive tool for assessing digital transformation in developing countries at all government levels. This framework uses qualitative and quantitative analysis to identify the strengths and weaknesses of digital governance status and provide recommendations for future improvement strategies. Countries that have tried to use DGRA to assess digital government readiness are Myanmar, Senegal, Lebanon, Vietnam, Kyrgyzstan, and Uzbekistan [21].

The DGRA has not specifically been made to assess the digital readiness of land services. Its use for the land service will require modifications and adjustments to suit the context. However, no research comprehensively assesses the readiness for the digital transformation of land services in Indonesia. This knowledge is needed to guide the improvement of aspects related to land services. For this reason, it is necessary to evaluate the quality of data and existing land service business processes. This paper aims to develop an assessment tool for evaluating the digital transformation of land services in Indonesia, based on the World Bank's DGRA model, which is not specifically designed for land services.

## 2. Materials and Methods

### 2.1. Materials and Sample Data

This study's materials and sample data include the DGRA framework toolkit, land service standard procedures, technical specification and user needs, textual land datasets, and spatial land datasets. The materials and data used are presented in Table 1.

**Table 1.** The materials and data used.

| Materials and Data Used | Description |
|---|---|
| 1. DGRA Framework Toolkit | Available online from the World Bank Website [22] (Figures 1 and 5) |
| 2. Land Service Standard Procedures | a. Government Regulation No. 24/1997 on Land Registration [23]<br>b. The Head of National Land Agency Regulation No. 1/2010 [10]<br>c. The Head of National Land Agency Regulation No. 6/2013 [24]<br>d. Minister of ATR/BPN Regulation No. 4/2017 [25]<br>e. Minister of ATR/BPN Regulation No.17/2015 [26] |
| 3. Technical Specification and User Needs | Technical Guidelines for Implementing Land Services [23,27] |
| 4. Textual Land Datasets | Land Registration Database in the land service<br>(Danurejan District, Yogyakarta City, Indonesia: 3731 records) (Figure 2) |
| 5. Spatial Land Datasets | Working Map and GeoKKP Parcel Map of Danurejan District, Yogyakarta City, Indonesia (Figure 3a,b) |

2.1.1. The DGRA Framework Toolkit

The DGRA Framework Toolkit is a series of questionnaires related to digital transformation indicators in government services in an excel file. DGRA consists of four digital regime assessments: Digital Leadership, Digital Services, Human Resources, Digital Infrastructure, Government Business Continuity, and Digital Legislation and Regulation (Figure 1). The four digital regimes consist of nine core group foundations that are used as the basis for infrastructure development and digital governance services, namely (i) leadership and governance, (ii) user-centered design, (iii) public administration and change management, (iv) capabilities, culture and skills, (v) technology infrastructure, (vi) data infrastructure, strategy, and governance, (vii) cybersecurity, privacy and resilience, (viii) legislation and regulation, and (ix) digital ecosystems.

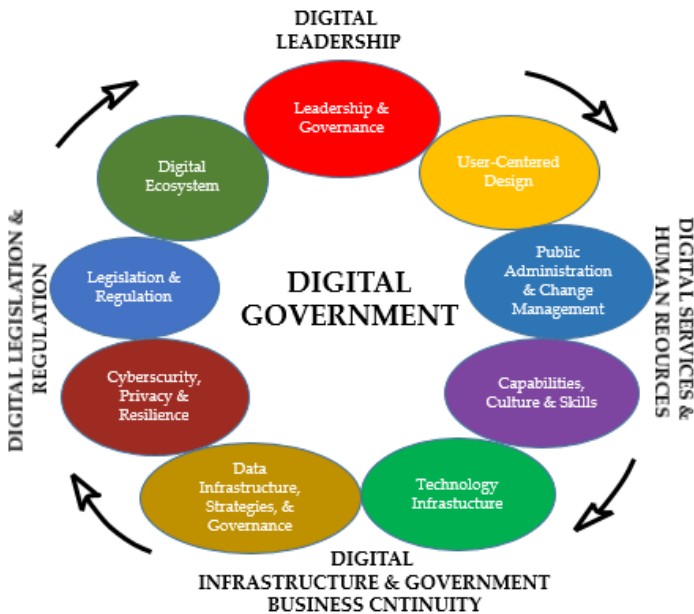

**Figure 1.** The nine core groups of the Digital Governance Assessment Framework (DGRA), redrawn from [21].

### 2.1.2. Land Service Standard Procedures

Land Service Standard Procedures is a series of regulations issued by the Republic of Indonesia's government at state and ministerial levels. They are Government Regulation No. 24/1997 on Land Registration [23]; The Head of National Land Agency Regulation No. 1/2010 on Standard Services and Land Regulation [10]; The Head of National Land Agency Regulation No. 6/2013 on Public Information Services in The National Land Agency of The Republic of Indonesia [24], Minister of ATR/BPN Regulation No. 4/2017 on Service Standards (ministerial levels) [25], Minister of ATR/BPN Regulation No.17/2015 on Land Service Standards in the Context of Investment [26].

### 2.1.3. Textual Land Datasets

The textual data used as a sample is taken from the existing database in the land service application (KKP). The sample's location is Danurejan District, Yogyakarta City, The Province of DIY, Indonesia (Figure 2). The Danurejan District consists of three villages, namely Suryatmajan, Tegal Panggung, and Bausasran (3731 records).

### 2.1.4. Spatial Land Datasets

The land spatial data in the form of land parcel maps consists of two types of maps (Figure 3), namely the working map (Figure 3a) and the land parcel map from the GeoKKP application (Figure 3b). The working map is a map used to place the land parcels, for which a certificate will be issued on the registration base map. This map is the initial product of the mapping process and is stored only on the land office's local server. If the land parcels' position has been mapped correctly on the working map, then the mapped parcel is uploaded to the Geo-KKP database (the central server). Not all land parcels mapped on the working map are uploaded to the GeoKKP database because there are still many land parcels in doubt regarding their location, shape, and area.

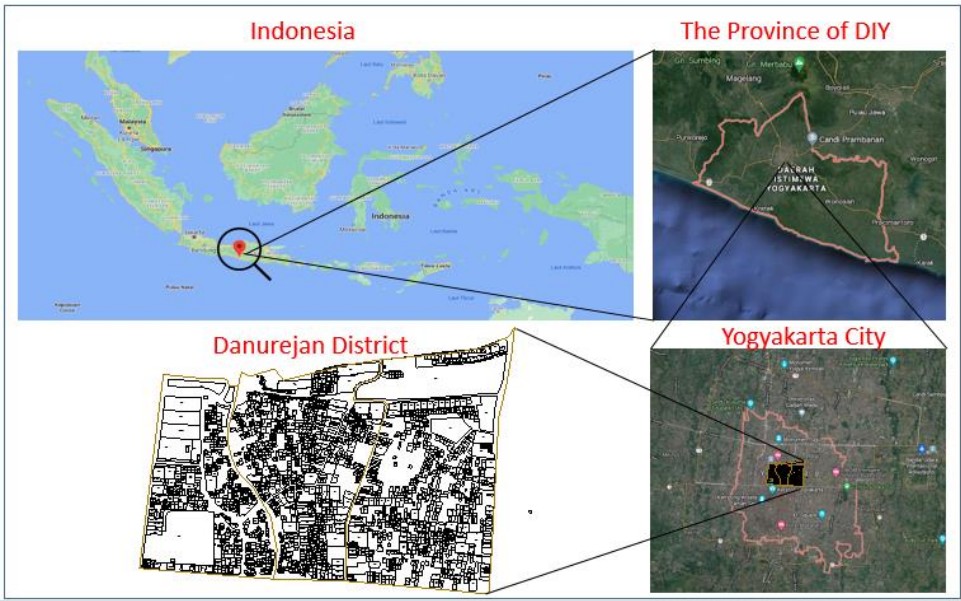

**Figure 2.** Location of the sample (Danurejan District, Yogyakarta City, Yogyakarta Special Province, Indonesia).

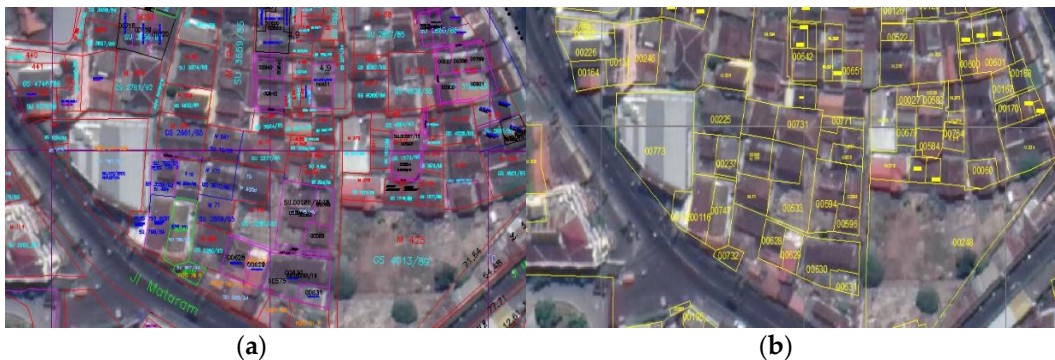

| (**a**) | (**b**) |

**Figure 3.** (**a**) Working map (the initial product of the mapping process); (**b**) GeoKKP parcel map (map of the final results of the plot mapping process, stored in the GeoKKP database on the central server).

### 2.2. Methods

The research workflow used in this research can be seen in Figure 4.

### 2.2.1. Desk Study and Interviews

The current land service business process in the Ministry of ATR/BPN was the main focus of this study. The land service business process can be seen from the standard procedures outlined in the form of regulations issued by the ATR/BPN. Desk studies were conducted to identify formal legislation and regulations that have been issued by the government. Desk studies were also conducted to find the technical specifications and user needs of the service process results. To evaluate how the current regulations on the land service business process are implemented, direct observations and in-depth interviews were conducted with stakeholders and structural officials at the central and city levels (Land Office). In-depth interviews were conducted with the Head of Planning Bureau of the Ministry of ATR/BPN, Head of Cooperation Section of the Ministry of ATR/BPN, Center for Data and Information of the Ministry of ATR/BPN, Head of Yogyakarta City Land Office, Head of Yogyakarta City Land Infrastructure Section, Land Deed Making Officials, and Land Service Applicants. Matters discussed in in-depth interviews were related to business processes and infrastructure, regulation, leadership, and human resources, as a primary part of assessing digital transformation readiness.

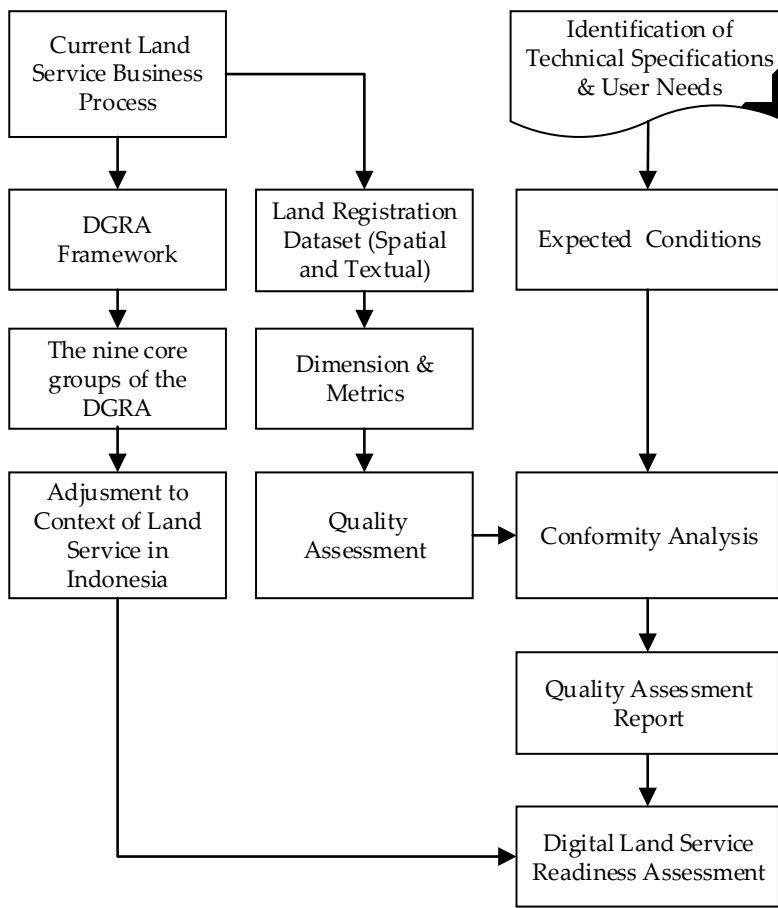

**Figure 4.** Research workflow.

2.2.2. Data Quality Assessment and Conformity Analysis

As a result of the land service business process, the land dataset must be assessed for its quality because it functions as an infrastructure for the continuity of the land service business process. The results of the quality assessment were used to confirm answers to questions related to these matters. The spatial and textual datasets taken from the land service application database (KKP) were used in this research.

The quality assessment was based on the specified dimensions (completeness, conformity, consistency, accuracy, duplication, and integrity). Based on these dimensions, the authors aimed to assess the quality of land datasets obtained from the Land Office of Yogyakarta City as a sample in this study. The dataset obtained was processed and analyzed using the IDE (Integrated Development Environment), R Studio computer program. The size of the data to be processed is getting bigger, so that improved data analysis tools/data science tools are needed. With R programming language, each selected dimension of data quality of the sample can be measured. The technical specification and user needs are then used as a comparison in conformity analysis.

The completeness assessment was carried out to assess which data were missing and unusable. The conformity assessment was carried out to assess which data were stored in a non-standard format. The consistency assessment was carried out to assess which data values gave conflicting information. The accuracy assessment was carried out to assess which data were incorrect or out of date. A duplicate assessment was carried out to assess which data records or attributes were repeated. An integrity assessment was carried out to assess which data were missing or not referenced. Regarding the assessment of the quality of spatial data, the authors used technical specifications stipulated in the technical

guidance documents published by the Ministry of ATR/BPN, mainly related to parcel areas' calculation.

The results of the quality assessment were compared with the technical specifications and identified user needs. A conformity analysis was carried out to compare the existing data quality assessment results with the expected data quality, to confirm the readiness assessment results based on the framework.

### 2.2.3. The Digital Land Service Readiness Assessment

We used the Digital Governance Assessment Framework (DGRA) [21] from the World Bank as a framework in this study. An adjustment to the context of land services was necessary (Figure 5). The adaptation is made by classifying the types of questions on the DGRA toolkits according to their level, namely at the head office (national level) and the land office (city-level). There is an adjustment in the use of language that makes it easier to interpret the questions. After the answers have been collected, the results are combined again in the original DGRA toolkit in Microsoft Excel.

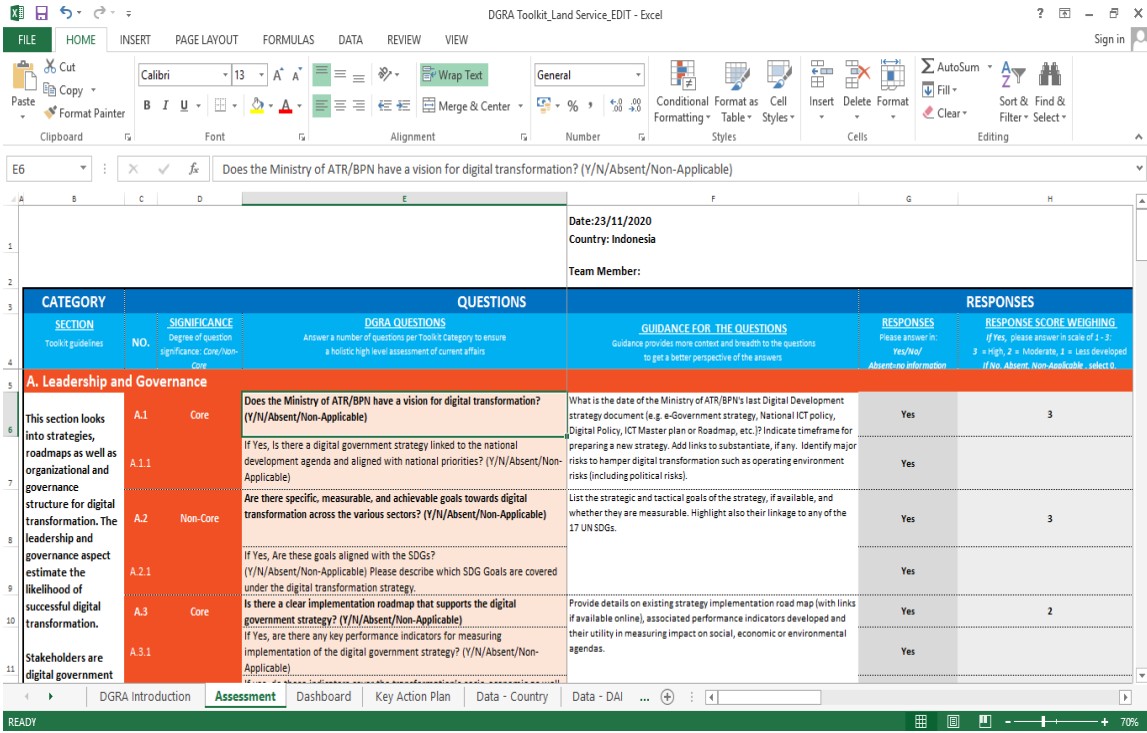

**Figure 5.** The DGRA for land services adapted from the World Bank DGRA Toolkit [21].

The DGRA consists of nine core indicators (Figure 1). These indicators were assessed using 67 questions. The questions were developed in excel format and given score weights. Score weighting on a scale of 1–3 provided a more gradient view of the extent of development, coverage, completeness of work, the maturity of capabilities, awareness levels, and the readiness of the platform, with 3 = highly developed and aware, 2 = moderately developed and aware, 1 = less developed and aware, and 0 = no, absent, or non-applicable. The answer to each question consists of four answers: yes = 0, no = 1, absent = 2, and N/A = 3. The results of the recapitulation of the assessment can be seen on the dashboard of the excel tools.

## 3. Results

### 3.1. The Result of Desk Study and Interviews

The results of this desk study and interviews can be elaborated descriptively. Regarding the results of interviews related to the land service business process at the Ministry of

ATR/BPN, it was found that digital transformation has become the primary vision. The implementation was still constrained by internal conditions, especially the documents and land data quality that need to be improved. The land office has started to convert documents from manual to digital. Efforts to improve land spatial and textual data quality have begun and are mainstreamed.

Indonesia's land service business process is based on Government Regulation No. 24/1997 on Land Registration [23] and several derivative regulations issued at the ministerial level. The regulation clearly states that the service business process must be simple, safe, affordable, up-to-date, and open. The Ministry of ATR/BPN has issued land service standards at the level of the Land Office (Regency), and at the level of the Regional Office (Province) [10], as well as at the level of the Ministry (Central) Office [25]. Then, specifically, the Ministry of ATR/BPN issued regulations on land service standards in investment activities [26]. Land service standards in these regulations include groups and types of services, service requirements, service fees, service completion times, service procedures, and reporting of service results involving actors, input, process, and output of land services.

The land service business process has been described in detail in an Activity Diagram for each type of service in the attachment section of land service standards. In the service process, the input, process, and output are all carried out by the Land Office. The applicant must collect the required documents from other government institutions or officials/partners who have the authority to issue the required documents, which are determined depending on the type of service desired, due to conditions in which documents related to land and property are not yet connected.

The applicant submits the required documents at the Land Office counter. The counter clerk checks the completeness of the documents by comparing their compliance with the stipulated requirements. If the requirements are incomplete, the files are returned to the applicant to be completed first and then returned to the counter. If the required documents are considered complete, the application file will be inputted into the Land Office Computerized system (KKP) without validation and cross-checking to the issuing institution against the applicant's documents' authenticity. The applicant will be given proof of receipt of the files. In short, the documents are not validated before entering the service system.

The required documents submitted to the Land Office counters are in the form of paper/manual, which still needs to be digitized and input by the input officer. Human error may occur during digitization. Digitalization was done incompletely. If synchronization and consistency are not monitored in the digitization process, there are still opportunities for inefficiency.

Next, the physical files and digital files will move according to the flow of service process types. Two service flows move parallel: the digital service flow in the KKP system and the manual service flow (delivery paper). There is often an inconsistency between the physical and digital files; the physical file may move first, or vice versa. Sometimes, during this physical file flow journey, one/several required documents are lost in an application file, which causes the service process to stop. The cause is sometimes due to the officers' negligence, or there could also be an element of intent such that the service process does not continue. The existing digital flow system has not been able to change employee behavior. The existence of a digital file flow (KKP) as a back-up has not yet become the primary reference/source in the service process. Quality control is carried out by officials at each level manually; the quality control results will vary depending on leadership, skills, and commitment from the staff on duty. Sometimes, this quality control process also involves other actors outside the land office officials, such as the village head, who is also involved in controlling the juridical file's correctness being processed.

Metadata management is an important matter that must be considered and developed in the land service business process at the Ministry of ATR/BPN. Metadata is information that describes the data [28,29]. Metadata informed by data producers include quantitative and qualitative data sources, including data collection methods, data representativeness,

curation, and statistical accuracy [30]. Land registration data should be equipped with metadata. The data source can be easily identified to confirm that the data were obtained with relevant criteria and classified based on its source. The location information can facilitate interoperability, digital identification, archiving, and preservation [28]. There is support for quality management. The service output is not a product, which is an absolute truth. If someone sues the service output with strong evidence and is supported by the court, the service product will be invalidated. Sometimes this happens when the conditions in the field are not the same as the current legal data, the conditions on the map, the condition of the documents, and the conditions of electronic data stored in the land office's computerized system. It still takes a very complex effort to synchronize these conditions. The principles of simple, safe, affordable, up-to-date, and open land registration have not been completely fulfilled.

However, efforts to improve the service system by utilizing the latest information technology have started to be implemented, such as Android-based mobile applications for data collection (Survey Tanahku) [31]. Allowing land service applicants to check service progress (Sentuh Tanahku) [32], and mobile and desktop E-Office apps [33] (an application that facilitates office administration). Recent developments at the Ministry of ATR/BPN have also launched a web-based service application that is more responsive to various devices: the KKP-Web version 2.0. [34]. Lots of other Android-based mobile applications have been developed to support tasks in specific fields. The recent land service application results from evolution, which provides an overview of the Land Information System's development in the Ministry of ATR/BPN. This development is triggered by the community's social, economic, and cultural needs (societal pull). This development is also encouraged by the development of information technology (technological push) [35,36]. Studies on utilizing the latest information technology have begun to be initiated; for example, studies on the possibility of implementing Blockchain technology. Awareness of quality culture has begun to become mainstream at the policy-making level. Additionally, the Public-Private Partnership (PPP) scheme for land service business processes has been initiated.

### 3.2. The Result of the Quality Assessment and Conformity Analysis

A quality assessment was carried out on land registration data at the Land Office of Yogyakarta City to get a more in-depth perspective on current conditions regarding the land data sample's quality. An overview of the data quality assessment is described in Table 2.

**Table 2.** The quality assessment results.

| Quality Dimensions | Quality Assessment Result |
| --- | --- |
| Completeness | Total records: 3731, incomplete records: NIB: 1185 (31.76%); SU Number: 613 (16.42%); left Number: 607 (16.26%); Publication Year: 3112 (83.40%) |
| Conformity | Non-conformity: in the location column, all records are stored in a non-standard format |
| Consistency | Inconsistency: the date was written in an inconsistent format |
| Accuracy | Inaccuracy: 1 record in which the writing year of the SU was wrong |
| Duplication | Duplication: 2 records duplication of the NIB |
| Integrity | Non Integrity: 1 record in which the writing year of the SU was wrong |

The textual data quality assessment results showed that, on the completeness dimension, many entries should have been filled in but were empty. These results were mostly the case for the fields regarding the parcel identification number (NIB), measuring letter number (SU), land right number, and year of publication. Of the total 3731 records in the Danurejan District, in the NIB column, 1185 records were NA (empty), in the SU number column 613 records were NA (empty), in the right number column 607 records were NA (empty), and in the year column, 3112 records were NA (empty).

The conformity assessment found that by filling in the location column for the category column, city column, sub-district column, and village column, the data is stored in a non-

standard format (string format). Based on regulations [37], the location must be in a numeric format. For each province, city, sub-district, and village in Indonesia, a unique code has been assigned. The dataset was Suryatmajan Village, Danurejan District, Yogyakarta City, Yogyakarta Special Region Province. Therefore, the writing of the area code for Suryatmajan Village is 3471041001: the first two digits, 34, are the code for the Special Region of Yogyakarta; the first four digits, 3471, is the code for the City of Yogyakarta; and the first six digits, 347104, is the code for Danurejan District. The last four digits are for village codes. Non-conformity was also found in recording time data with different formats.

In assessing the accuracy and duplication dimensions, it was found that there were data records of two fields in the same village in which different parcels were given the same NIB, with validated SU status. This recording was considered inaccurate, and there was a duplication of the NIB. Another inaccuracy found was a data record in which the SU's writing year was wrong. In the consistency assessment, it was found that the date was written in an inconsistent format.

Regarding the spatial data quality assessment, we used the technical specifications that have been stipulated in the technical guidance documents published by the Ministry of ATR/BPN. One of the technical guidelines' provisions in the area difference between the area calculated on the parcel map and the area stated in the SU document; the tolerance for difference is less than or equal to 5%. From the assessment results, it was found that from a total of 3731 records, 418 parcels (11.20%) had a difference of more than 5%. Of the 418 parcels with an area difference of more than 5%, there was a difference between 5.02% and 1677%. From the visual analysis, it was also found that there were parcels mapped outside the village administrative boundaries due to the wrong mapping location being used.

### 3.3. The Result of the Digital Land Service Readiness Assessment

The readiness assessment results using the DGRA framework [21] adapted to Indonesia's land services can be seen in Table 3 and Figure 6.

**Table 3.** The DGRA results for land services in the Ministry of Agrarian and Spatial Planning/National Land Agency (ATR/BPN).

| No | Section | Total Score | Yes | No | Absent | N/A |
|----|---------|-------------|-----|-----|--------|-----|
| 1 | Leadership and Governance | 2.7 | 9 | 0 | 0 | 0 |
| 2 | User-Centered Design | 2.0 | 8 | 0 | 0 | 0 |
| 3 | Public Administration Reforms and Change Management | 2.3 | 6 | 0 | 0 | 0 |
| 4 | Capabilities, Culture, and Skills | 1.9 | 7 | 0 | 0 | 0 |
| 5 | Technology Infrastructure | 1.7 | 5 | 5 | 0 | 0 |
| 6 | Data Infrastructure, Strategies, and Governance | 1.8 | 5 | 3 | 0 | 0 |
| 7 | Cyber Security, Privacy and Resilience | 1.0 | 4 | 0 | 0 | 0 |
| 8 | Legislation and Regulation | 1.4 | 5 | 3 | 0 | 0 |
| 9 | Digital Ecosystem | 1.7 | 7 | 0 | 0 | 0 |

Table 3 and Figure 6 show that three areas score above two, namely leadership and governance (2.7), user-centered design (2.0), and public administration reforms and change management (2.3). The other areas score below two, namely capabilities, culture and skills (1.9), technology infrastructure (1.7), data infrastructure, strategies, and governance (1.8), cybersecurity, privacy and resilience (1.0), legislation and regulation (1.4), and digital ecosystem (1.7).

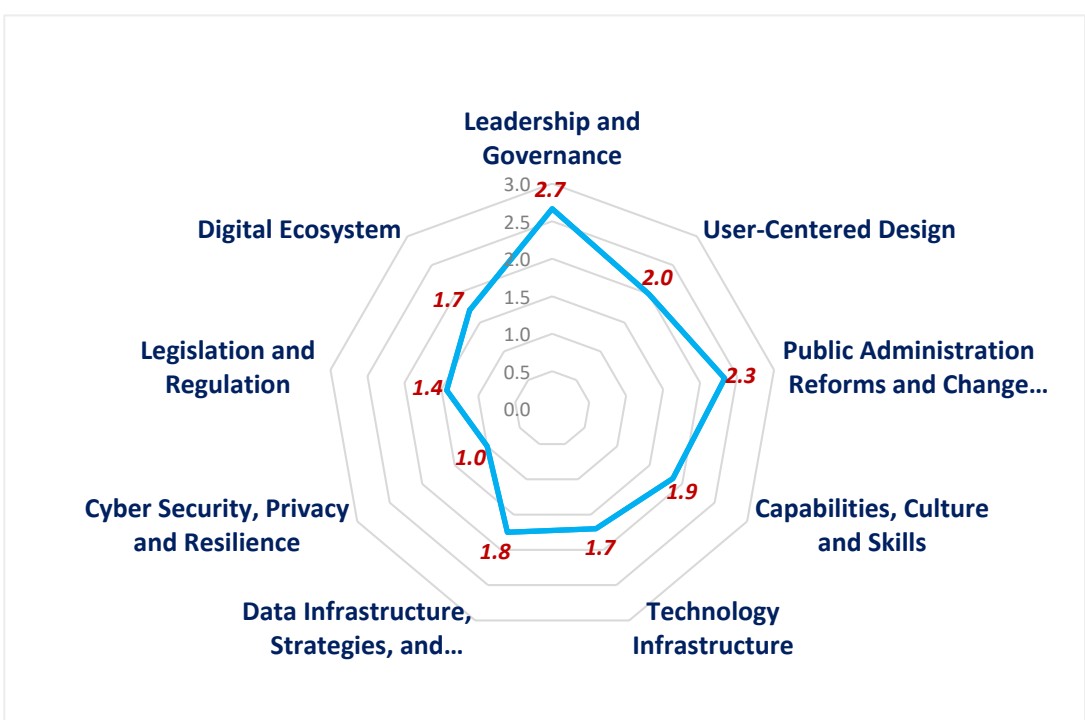

**Figure 6.** The digital government readiness score per section for land services in the Ministry of ATR/BPN.

It can be seen that leadership and governance obtained the best score of 2.7. In this section, there is a roadmap for implementation. The policy of the digital government, the technical ability of the Ministry of ATR/BPN's staff to implement digital transformation strategies, internal learning at all levels regarding digital skills, collaboration among service staff on specific themes and projects, knowledge-based implementation group in digital technology, and several other matters related to leadership and governance, which tended to have an answer of "yes," with most of them having a "moderate" weight. If observed in the field related to human resource development at the Ministry of ATR/BPN, it is clear that there has been a change in an encouraging direction. Human resource development activities (education and training) and competency assessments have begun to apply online learning technology, blended learning, distance learning, and computer-based tests [38].

Meanwhile, the lowest score was in the cybersecurity, privacy, and resilience section, with a score of 1.0. In this section, we assessed the availability of a cybersecurity strategy and policy document; the availability of a cybersecurity unit or center within a core entity to manage and maintain the security of all digital assets and platforms; the existence of a Computer Emergency Response Team (CERT); collaboration with regional and international governments or organizations to share information on and mitigate cyber threats or risks; and whether there is a National Critical Infrastructure Protection Plan. Even though all the answers to these questions were "Yes," the weight of these answers was "1" or "less developed." Matters related to cybersecurity, privacy, and resilience need attention immediately to reach the required capacity. Technology infrastructure, legislation, and regulation also require further attention and data infrastructure, strategies, and governance. Many of the questions related to these matters received "No" answers (Figure 7), leading to a low average score.

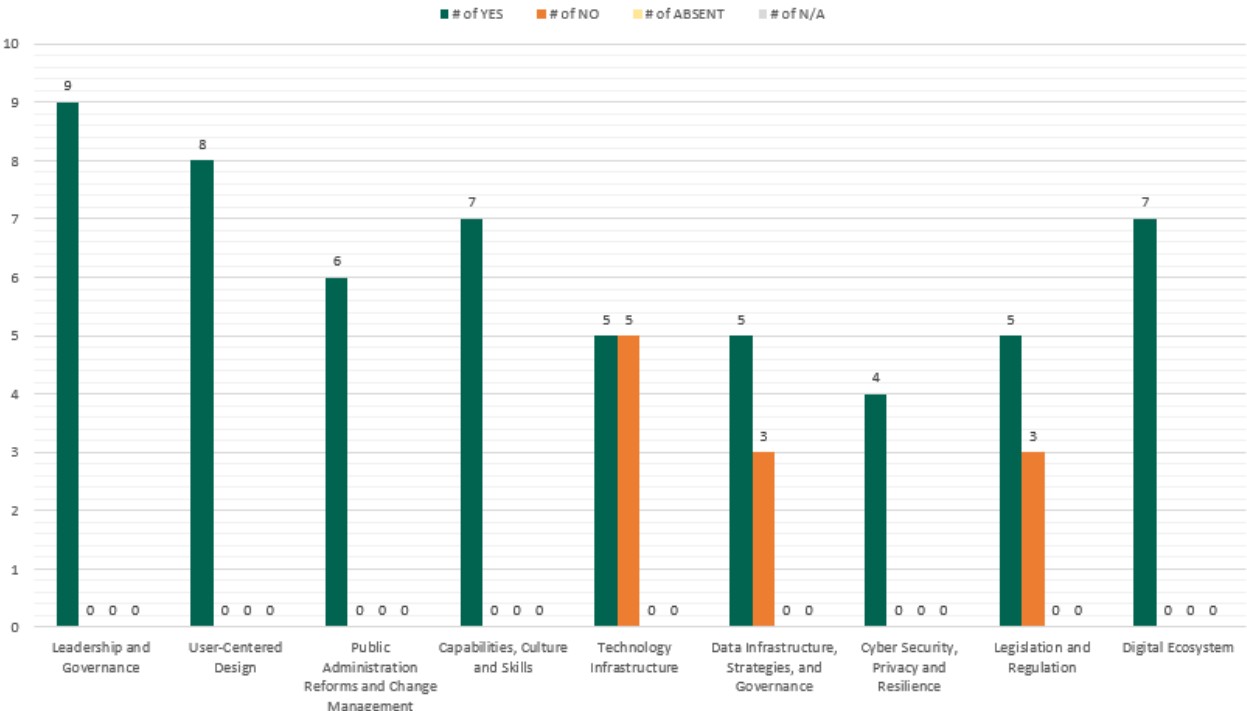

**Figure 7.** The DGRA results per section for land services in the Ministry of ATR/BPN.

## 4. Discussion

The strategic goal of the Ministry of ATR/BPN by 2025 is the realization of a world-standard institution. This paper seeks to assess the internal readiness to implement a digital transformation roadmap to achieve its vision. The readiness assessment results are presented in detail in Table 3 and Figure 6, with readiness scores for the nine core groups used as the basis for infrastructure development and digital governance services quantitatively adapted from the DGRA framework from the World Bank [21].

This study has successfully implemented a comparative approach in quality assessment. The comparative approach here is the conformity analysis of the existing data quality (quality metrics) with the expected data quality. This approach differs from the comparative approach introduced by Pipino et al. [39], which compares data quality metrics and survey data quality. The analysis compares data collected from the survey (stakeholder perceptions) and the results from quantitative metrics. The resulting comparisons are used to diagnose and prioritize critical areas for improvement. This approach is often referred to as a diagnostic approach because of its diagnostic nature. Whereas in this study, a comparative approach was carried out by comparing the quality metrics with the expected data quality. The expected data quality is from direct observations, in-depth interviews, and by identifying technical specifications and standards issued by the Ministry of ATR/BPN as a producer and a data user.

The quality assessment was used to confirm the results of the readiness assessment. Quality is defined as the totality of a product's characteristics with its ability to meet express or implied needs [40] and called 'suitability for intended use' [41–44]. It further encompasses conformance to requirements [45] or user satisfaction [46]. High-quality data is a valuable asset that can increase user satisfaction; high-quality data can increase revenue and profit and be a strategic competitive advantage [47]. Good data quality becomes the basis for better planning and decision-making and improving the quality of public services. An organization that ignores data quality can experience major problems and losses [43]. The use of inappropriate quality data also results in substantial social and economic impacts on data users [42]. Data has become a complex and multi-dimensional entity with advances in information technology [48]. Dimensions of data need to be determined

to be better understood, measured, and improved in quality. The dimension of data quality is a single aspect or property of data [42] that can be measured and improved. According to Sally McCormack [49], the dimensions used to measure data quality are completeness, conformity, consistency, accuracy, duplication, and integrity. The sample dataset's quality assessment results related to the completeness, conformity, consistency, accuracy, duplication, and integrity show a need to improve the quality of land data used in digital land services.

As we compare this research with previously published studies [11,12,20], we could see that this study more specifically assesses the readiness for digital transformation in land services by adapting the DGRA framework. The previous study [11,12] built a framework based on a synthesis of several other frameworks for assessing land services' performance and analyzing the usability of implementing full digital land registration [20]. Another difference is the indicators used in the assessment. The nine core groups of the DGRA Framework were used in this study, while the assessment indicators used in previous studies [11,12] were a combination of indicators from several synthesized frameworks.

The digital readiness assessment should consider the ten principles of the DGRA itself [21]. The readiness assessment of land services digital transformation in the context of Indonesia carried out in this study has not fulfilled the ten principles as a whole. Still, it has shown quantitative readiness figures from each category assessed to be developed or improved towards ideal conditions. The readiness assessment of digital transformation in Indonesia's land services, with complete DGRA principles, should be fulfilled in further research. In this regard, this examination is unique in that no exploration has been led in Indonesia that utilizes this kind of assessment model. It is an experimental investigation on readiness assessment utilizing essential information based on direct observation. Henceforth, it can fill in as writing to mainstream researchers.

This paper has demonstrated that the DGRA Framework can be applied to assess digital transformation readiness at all levels and sectors of governance, particularly in land services in developing countries such as Indonesia. In this case, adjustments and adaptations of the framework are needed, and they have been carried out in the readiness assessment process to suit Indonesia's land services. The assessment results should become valuable input in the government's efforts to improve digital land services since they show core areas that require urgent improvements.

What must be considered are the institutional challenges and the estimated impact of digital transformation on land services. Recently, Ukraine proposed a new methodology that focuses on the impact of digital transformation by identifying core governance [50]. This method's concept consists of three main things, namely the priority of institutional reform and governance from operational learning, twelve principles of the OECD Digital Government Toolkit [51], and the 2016 World Development Report on Digital Dividends [52].

In this research, the sample dataset used to assess the quality of the product/data are currently obtained only from one land office, considering that the rules and standard operating procedures for services are the same for all Indonesian regions. In reality, the conditions of the dataset in each land office may vary. Therefore, in order to have a complete figure of land services in Indonesia, it will be necessary to add samples of data from other land offices.

## 5. Conclusions

This study has developed a readiness assessment tool for the digital transformation of land services. Indonesia has begun to use digital applications in land services and land registration activities. However, until recently, this has not been the main regime in the service process. The research's main idea is the challenge to ensure that all types of land services can be implemented digitally. The quality assessment of land services using the sample dataset has provided support for assessing digital services' readiness, especially in developing a strategy for the continuity of the land service business process. To be ready, the land office must carry out many very complicated processes to improve land

registration data quality. Using the DGRA toolkit, the assessment results indicate that the readiness level still needs improvement, especially in the core parts related to cyber-security, privacy, and resilience. Even though the leadership and governance section shows a reasonably high score, other core sections, namely technology infrastructure, legislation and regulation, and data infrastructure, strategies, and governance, are mediocre, and therefore they need improvement.

The study shows that applying the DGRA framework from the World Bank to assess the land services provides more information beyond technicalities. The framework covers broader areas, such as Leadership and Governance, Public Administration Reforms, and Change Management, which will be valuable for improving land services in Indonesia.

This study has successfully assessed each indicator on nine core groups of the DGRA Model related to digital-based land services readiness in Indonesia. However, by considering the vast area of Indonesia and its varying conditions, further research concerning the use of the DGRA model needs to be done by including more land offices to get more representative data samples and observations.

**Author Contributions:** Conceptualization, T.A. and K.K.; methodology, T.A., D.D., and S.S.; formal analysis, K.K., and T.A.; investigation, T.A.; writing—original draft preparation, K.K.; writing—review and editing, T.A. and D.D. and S.S.; visualization, K.K.; supervision, S.S.; project administration, K.K.; funding acquisition, T.A. and K.K. All authors have read and agreed to the published version of the manuscript.

**Funding:** This research was funded by the Directorate of Research Universitas Gadjah Mada. Doctoral program scholarship awarded by Sekolah Tinggi Pertanahan Nasional.

**Institutional Review Board Statement:** Ethical review and approval were waived for this study due to data presented is not related to any case details, personal information, and/or images of respondents and follows the applicable regulations in Indonesia.

**Informed Consent Statement:** Informed consent was obtained from all subjects involved in the study.

**Data Availability Statement:** The data presented in this study are available on request from the corresponding author. The data are not publicly available due to government regulations in Indonesia.

**Acknowledgments:** The authors would like to thank Eko Suharto, Wisang Wisudanar, Sidiq Isnanto, Reza Abdullah, Suwandi, Ketut Ari Sucaya, and Gabriel Triwibawa for responding to the in-depth interviews. The authors would also like to thank Krisostomus Nova, who helped with data analysis. Fahmi Charis Mustofa, Arie Yulfa acted as discussion partners, M. Nazir Salim for the similarity check tools, Purwoko for the English editing tools.

**Conflicts of Interest:** The authors declare no conflict of interest.

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
