# Peer review of "Digital Transformation of Land Services in Indonesia: A Readiness Assessment"

_land, doi:10.3390/land10020120_

Round 1

Reviewer 1 Report

I have now reviewed the manuscript by Kusmiarto et al., to evaluate the readiness of a land office to achieve digital transformation visions. The significance of research and objectives are not clearly stated, but apparently this was not done before in this area, which is the main reason for this exercise. This paper would certainly be of interest to readers of this journal.  While this is certainly a noble effort, however there are several flaws in this study that must be addressed.

Major Comments

  • In the introduction, the purpose of the research and significance should be stated. The aim of the paper should be clearly stated in one sentence and presented in the introduction. Also, the main idea, importance, novelty, etc. Can be indicated in this section
  • There is insufficient novelty, in the sense that no new methodology is presented.
  • In all, the manuscript has the appearance of applying some well-known techniques without a proper critical and scientific attitude. There is too limited description of the actual methodology and in particular it is not made clear what the innovative aspects of your work are.
  • The paper does not contain a scientific discussion, which is essential to properly understand the consequences of your research.
  • Comparisons with other studies have to be provided in the discussion section
  • Literature review part is needed to be elaborated more; focus should be given as per land administration services. The theoretical study should include more recently published papers from high-level scientific journals. It is highly recommended to extend the literature review, including scientific papers published in highly ranked journals indexed in the Clarivate Analytics Web of Science and Scopus databases
  • The references cited in the text is not as per the journal criteria, see the journal’s instructions for authors for details on style
  • The whole Conclusions section just seems to be a self-congratulatory repetition of information from elsewhere without actually concluding very much.
  • Recommendations for future studies are not significant as per the findng of this study, so rewrite the future studies
  • Keywords should not be the repetitions of the title words, please find such words which are not in the title, this way search engines of the web will find your manuscript with higher probability.
  • The references list is not formatted according to journal criteria; References: see the journal’s instructions for authors for details on style
  • Please provide research limitations

Author Response

Dear Reviewer,

Best Regards,

Kusmiarto

Reviewer 2 Report

This paper is about land transformation assessment done so that the government in Indonesia can have some recommendations. There is no major flaw in the paper. The study performed is obvious, but it has great significance. Authors have kept their focus on the issue and did not deviate unnecessarily. However, they can still make the paper better by having professional-looking high-quality illustrations and diagrams. Not a big issue, but authors must proofread the paper again, typically for grammar.

Author Response

Dear Reviewer,

Best Regards,

Kusmiarto

Reviewer 3 Report

It is a study on the governance of the land, the authors are committed to promoting the digitization of the land to favor economic development in Indonesia, by increasing the gross domestic product.

In the introduction the objectives are clear.

However, the introduction is very poor in references on the digitization of the land in other countries.

In material and methods it is suggested to include a map on the location of the study in Indonesia.

In my opinion, the results section should be rewritten, since there are questions that should go to material and methods; For example, it talks about the use of 193 data on line 192, but does not mention them in methodology, it is suggested to make a table with the data used.

In the discussion, you could make a comparative analysis with the digitization of the land in other countries, and compare the increase in gross domestic product between countries with and without digitization of the land.

The conclusions are poor, we advise enriching this section.

Author Response

Dear Reviewer,

Best Regards,

Kusmiarto

Round 2

Reviewer 1 Report

Thank you for addressing my comments and revising the manuscript. The current discussion section, mostly repeats the results and reviews of literature without any indication of, for example, why the study outperformed the others. Please rework on discussion sections and resubmit. Good Luck

Author Response

Dear Reviewer,

Best Regards,

Reviewer 3 Report

The article has been improved and can be published

Author Response

Dear Reviewer,

Best Regards
